# An oral cholera vaccine in the prevention and/or treatment of inflammatory bowel disease

**Marine Meunier** [1]*, **Adrian Spillmann**[2], **Christel Rousseaux**[3], **Klaus Schwamborn**[1], **Melissa Hanson**[1]

**1** VALNEVA SE, Saint-Herblain, France, **2** VALNEVA AUSTRIA GMBH, Vienna, Austria, **3** Intestinal Biotech Development, Faculté de Médecine—Pole Recherche, Lille, France

☺ These authors contributed equally to this work.
* marine.meunier@valneva.com

## Abstract

The oral cholera vaccine WC-rBS consists of 4 different inactivated strains of *Vibrio cholerae* (LPS source) admixed with recombinant cholera toxin B subunit. Because of its unique composition and anti-inflammatory properties reported for both CTB and low doses of LPS from other Gram-negative bacteria, we speculated that WC-rBS might have anti-inflammatory potential in a chronic autoimmune disease such as inflammatory bowel diseases. First *in vitro* endotoxin tolerance experiments showed the surprising WC-rBS potential in the modulation of inflammatory responses on both PBMCs and THP1 cells. WC-rBS was further evaluated in the Dextran Sodium Sulfate colitis mouse model. Administrated orally at different dosages, WC-rBS vaccine was safe and showed immunomodulatory properties when administered in a preventive mode (before and during the induction of DSS colitis) as well as in a curative mode (after colitis induction); with improvement of disease activity index (from 27 to 73%) and histological score (from 65 to 88%). Interestingly, the highest therapeutic effect of WC-rBS vaccine was observed with the lowest dosage, showing even better anti-inflammatory properties than mesalamine; an approved 5-aminosalicylic acid drug for treating IBD patients. In summary, this is the first time that a prophylactic medicine, safe and approved for prevention of an infectious disease, showed a benefit in an inflammatory bowel disease model, potentially offering a novel therapeutic modality for IBD patients.

## Introduction

Inflammatory bowel disease (IBD) is an autoimmune disease characterized by the loss of immune tolerance for normal bacteria present in the gut microbiome. Therefore, the immune system of IBD patients attacks bacteria and induces chronic inflammation, which has been linked to increased cancer risk [1]. The global prevalence of IBD has been increasing since 2000, and IBD now affects up to 1 in 200 individuals in Western countries [2]. IBD encompasses two distinct disorders, Crohn's disease (CD) and ulcerative colitis (UC), which differ in pathophysiology, affected parts of the gastrointestinal (GI) tract, symptoms, complications,

**Data Availability Statement:** All relevant data for this study are within the paper and its Supporting Information files.

**Funding:** The author(s) received no specific funding for this work

**Competing interests:** Marine Meunier, Adrian Spillmann, Klaus Schwamborn and Melissa Hanson were all employees of the Valneva group when the work was done. The Swedish subsidiary of the Valneva group owns Dukoral®, i.e. the WC-rBS vaccine. This does not alter our adherence to PLOS ONE policies on sharing data and materials.

disease course and management. CD is characterized by discontinuous intestinal lesions anywhere in the GI tract, and involves chronic, relapsing transmural inflammation that can lead to chronic abdominal pain, diarrhea, obstruction and/or perianal lesions. UC affects only the colon, the lesions are continuous, and inflammation is superficial, which can lead to erosions, ulcers, and bloody diarrhea.

Depending on the severity of the disease, different drugs can be administered for the treatment of IBD such as corticosteroids, aminosalicylates, biologics, supportive medications, and immunosuppressive drugs. Especially, mesalamine is a safe and well tolerated aminosalicylate anti-inflammatory drug widely used as a first-line therapy in IDB patients [3]. Despite multiple options available for IBD patients [4]–considering the strong side effects and high relapse rates for current therapies–there remains a large unmet need to improve IBD outcomes with safe and effective treatments.

The WC-rBS vaccine (Dukoral®, Valneva, Solna, Sweden) is an orally administered vaccine for the prevention of cholera and in some countries also indicated to help prevent diarrhea caused by heat-labile toxin-producing enterotoxigenic *E. coli* (LT-producing ETEC). It consists of 4 different inactivated strains of *Vibrio cholerae* (*V. cholerae*) admixed with recombinant cholera toxin B subunit (rCTB). CTB is part of the cholera toxin (CT) secreted by the bacterium *V. cholerae*. In its pentameric form, it is responsible for the toxin binding to intestinal epithelial cells via ganglioside GM1. The monomeric A subunit (CTA) is responsible of the secretory effects of the toxin leading to diarrhea. CTB can be produced recombinantly (henceforth referred to as rCTB) or purified from whole cholera toxin (henceforth referred to as CTB).

The immunomodulatory properties of CTB/rCTB have been reported in several studies for different diseases [5–7]. *In vitro*, Burkart *et al* [8] demonstrated that pre-exposure of monocytes (Mono Mac 6 cells) or PBMCs to CTB prevented a proinflammatory reaction to a subsequent LPS challenge with significantly lower levels of secreted pro-inflammatory TNFα, IL6, IL12 cytokines and transient elevated secretion levels of the anti-inflammatory cytokine IL-10. In the context of Behcet's disease (an inflammatory disorder leading to uveitis as a major complication), Phipps *et al* indicated the induction of uveitis in Lewis rats when orally administered with an HSP60-derived peptide; whereas administration of a fusion protein of rCTB linked to an HSP60-derived peptide prevented uveitis development. This indicated a tolerogenic role of rCTB [9]. In a subsequent human clinical trial, three oral administrations per week of the rCTB fusion protein over 12 to 16 weeks prevented uveitis relapse in Behcet's disease patients [10]. Another clinical study in Crohn's disease patients showed safety and tolerability of rCTB. When administered orally 3 times weekly over two weeks, rCTB significantly decreased disease activity index in 40% of patients [11].

In addition to the immunomodulatory properties of CTB, low doses of the lipopolysaccharide (LPS) of Gram-negative bacteria can also induce anti-inflammatory and tolerogenic responses. In particular, Mendes *et al* highlighted differential regulation of TLR signaling pathways when human PBMCs were pretreated with low dose *E. coli* LPS before being challenged with high dose *E. coli* LPS [12]. Low dose *E. coli* LPS also modulated secretion of inflammatory mediators such as the upregulation of IL-10 and the downregulation of TNFα or IL-12 [12]. *E. coli* LPS pretreatment has also been shown to drive polarization of M2-macrophages which act to enhance wound healing and control the inflammatory response via the downregulation of TNFα, cyclooxygenase-2 and tissue factor [13]. *In vivo*, orally administered low dose *E. coli* LPS is protective against sepsis in mice via modulation of pro-inflammatory responses induced by cecal ligation and puncture [14]. Although not a purified source of LPS, inactivated cholera strains (a rich source of *V. cholerae* LPS) also promote anti-inflammatory responses in endotoxin tolerance experiments, with reduced TNFα and increased IL-10 secretions [15].

More recently, the combination of CTB with low doses of *E. coli* LPS was reported to enhance anti-inflammatory properties. *In vitro* experiments on healthy human monocyte derived dendritic cells indicated rCTB fused to proinsulin (a type 1 diabetes autoantigen) modestly upregulated the secretion of IL-10 and indoleamine 2, 3-dioxygenase (IDO1) and drove DCs to a tolerogenic state [16]. However, in this study, the presence of residual *E. coli* LPS in partially purified CTB fusion protein markedly enhanced CTB's anti-inflammatory properties resulting in high levels of secreted IL-10 and IDO1 by DCs.

In the context of IBD, an engineered rCTB containing the KDEL endoplasmic reticulum retention motif was reported to promote mucosal wound healing *in vitro* on a Caco-2 human colonic epithelial model via the secretion of TGF-β [17]. Consistently, *in vivo* oral administration of rCTB-KDEL in a Dextran Sodium Sulfate (DSS) acute colitis mouse model protected against colon mucosal damages and decreased local inflammation level, histopathological score, and disease activity index [17, 18]. In addition, the therapeutic effect of rCTB-KDEL was confirmed in a chronic DSS model. Indeed, weekly oral administration of rCTB-KDEL reduced DAI, inflammation, and histological markers of chronic colitis [18].

Due to the WC-rBS vaccine's specific composition of highly purified rCTB combined with *V. cholera* strains–which is a rich source of LPS albeit structurally different from *E. coli* LPS–we speculated WC-rBS may have anti-inflammatory potential in a chronic autoimmune disease such as inflammatory bowel diseases (IBD). The goal of this study was to investigate the potential beneficial properties of WC-rBS vaccine in autoimmune disease. *In vitro* endotoxin challenge experiments on both PBMC and THP1 monocytes revealed the ability of WC-rBS to modulate inflammation. The vaccine was then tested *in vivo* in the DSS colitis model, which is widely used to induce epithelial damage and subsequent colitis in mice. WC-rBS vaccine administration induced strong improvement of disease outcome on both the macro- and tissue-level scales. Interestingly, the lowest tested dosage presented the greatest beneficial effect, superior to the approved drug mesalamine (Pentasa® Ferring, Gentilly, France). To our knowledge, this is the first report of the therapeutic effect of WC-rBS–a prophylactic oral cholera vaccine–for the treatment and prevention of intestinal colitis and highlights its potential to be repurposed in the context of IBD and more broadly in autoimmune diseases.

## Materiel and methods

### THP-1 cell culture

THP-1 cells were obtained from ATCC® (TIB-202™) and cultured following ATCC recommendations at 37°C and 5% $CO_2$ in RPMI 1640 medium (Life Technologies, A10491-01) containing 0.05 mM 2-mercaptoethanol (Sigma, M3148), 10% fetal bovine serum (SAFC, 12003C) and penicillin and streptomycin antibiotics (Lonza, DE17-602E).

### PBMC isolation

Human PBMCs were isolated from freshly sampled blood from normal healthy donors (protocol adapted from Burkart *et al* [8]). Those blood samples were obtained from the French blood agency (Etablissement français du sang) and fully anonymized after informed consent from the donor obtained. Whole blood was centrifuged on a lymphocytes separation medium gradient (Eurobio—CMSMSL01-01) at 900 g for 25 min, followed by washing of the enriched cells in a PBS solution (Gibco, 10010–015) containing 1 mM EDTA (Gibco, 15575–038) and 2% FBS: one at 400 g for 8 minutes and two at 200 for 10 minutes. Cells were resuspended in RPMI 1640 (Sigma, R0883) supplemented with 10% FBS, 2 mM glutamine and penicillin and streptomycin antibiotics. In stimulation assays, isolated PBMCs were cultured at 37°C and 5% CO2 at a final concentration at $1x10^6$ cells/mL.

## Endotoxin tolerance experiments

Isolated PBMCs or THP1 cells were seeded in plates and pretreated with media, recombinant CTB (Valneva Sweden) or WC-rBS at a final cell density of $1x10^6$ cells per mL for PBMCs and $2x10^5$ cells per mL for THP1. After 16 to 24 hours incubation at 37˚C and 5% $CO_2$, cells were washed 3 times with fresh culture media and then challenged with a 1 μg/mL LPS solution (Sigma, L4391). After 6 to 24 additional hours of incubation, plates were centrifuged, and supernatants were collected for cytokines analyses.

## Cytokines analysis

TNFα and IL-10 cytokines concentrations in supernatants were determined by ELISA, using human DuoSet ELISA kits (R&D systems, DY210 and DY217B respectively), according to manufacturer recommendations. Briefly, after an overnight coating with capture antibody and a 1% BSA blocking step, appropriately diluted samples and standards were incubated on plates for 2 hours at room temperature. After washing, detection antibody and then a streptavidin solution were incubated in wells before a revelation step using a TMB solution (SeraCare, 5120–0047) and $H_2SO_4$ (LCH Chimie, MC3067691000) as a stop solution. Optical density reading was performed at 450 and 570 nm and cytokines concentrations determined by regression from standard curves.

## Animals

Nine-week-old C57BL/6 female mice were obtained from Charles River laboratories (Arbresle, France). All the studies were approved by the local investigational ethics review board (Nord-Pas-de-Calais CEEA N˚75, Lille, France) and French government (agreement n˚-APAFIS#7542-20 17030609233680)

## In vivo experiments

Two different preclinical experiments were performed. Ten mice per group, randomly assigned, were immunized with WC-rBS vaccine or PBS by oral gavage on days D-6, D-3, D0, D3 and D6 for WC-rBS evaluation as a preventive treatment; or on days D3 and D6 for evaluation as a curative treatment. WC-rBS was given at varying dosages as described in Table 1. To prepare dosages, dilutions of whole WC-rBS into phosphate buffer saline (PBS) buffer were performed, controlling the amount of rCTB within each WC-rBS dose. As a benchmark, one group of mice received mesalamine® (Ferring) ad libitum, mixed in food for the duration of the experiment.

For colitis induction, C57BL/6 mice received 2.5% of DSS (45kD; TDB Consultancy AB, Uppsala, Sweden, Ref DB001-43) in their drinking water for 5 days (from D0 to D5) followed by a regime of 7 days of regular water. In the C57BL/6 genetic background, mice experience peak of acute colitis 3 days after the last DSS administration, and chronic-like inflammation

**Table 1. WC-rBS dilutions.** The WC-rBS vaccine is composed of highly purified rCTB combined with inactivated V. cholera strains. WC-rBS was diluted in PBS for mouse studies. The amount of rCTB in μg and number of inactivated bacteria in CFU (coloning forming unit) and the related LPS content per WC-rBS dose are described here.

|  | rCTB (μg) | Inactivated bacteria (CFU) | LPS content (EU) |
|---|---|---|---|
| WC-rBS 25 | 25 | $2.9 \times 10^9$ | 17,95 |
| WC-rBS 5 | 5 | $5.8 \times 10^8$ | 3,59 |
| WC-rBS 3 | 3 | $3.5 \times 10^8$ | 2,15 |
| WC-rBS 1 | 1 | $1.2 \times 10^8$ | 0,72 |

occurs 7 days after the last DSS administration [19]. Each experiment also included a healthy control group of 5 unvaccinated and DSS untreated and unvaccinated mice.

Disease activity index was calculated daily in a blinded manner, based on the evolution of the body weight, stool consistency and the presence of occult blood as previously described [20]. At euthanasia on day D12, colon length was measured and global histological score was blindly assessed on colonic tissues stained with May-Grunwald-Giemsa and as described previously [21], both parameters reflecting the level of inflammation.

## MPO quantification

Myeloperoxidase (MPO) is an enzyme contained in polymorphonuclear neutrophil primary granules and constitutes an indirect marker of inflammation. MPO activity was quantified by ELISA (ref HK210-01, Cliniscience) in the colonic lesions at the end of the experiment. According to manufacturer's recommendations, colon specimens were homogenized with an Ultra Turrax T8 (Ika-Werke, Staufen, Germany) in a phosphate buffer (pH 6.0) containing 0.5% hexadecyltrimethyl ammonium and subjected to two sonication and freeze-thaw cycles. Suspensions were centrifuged at $14,000 \times g$ for 15 min at 4˚C and the supernatants were treated with o-dianisidine hydrochloride at 1 mg/mL and 0.0005% hydrogen peroxide. Optical density was read at 450 nm with a Versamax microplate reader (MDS Analytical Technologies, Saint-Grégoire, France) and MPO concentration determined by regression. Results were expressed in MPO amount (ng) per total quantity of proteins (mg) determined by the Bradford method.

## Statistics

For *in vitro* experiments, statistics were performed using one-way ANOVA with Dunnett's multiple comparison post-test using Graphpad Prism version 9.4 (Graphpad Software, La Jolla, CA, USA). For *in vivo* studies, all comparisons were analyzed using the Permutation Test for two independent samples using the StatXact software (Cytel Inc, Cambridge, MA, USA). Differences were considered statistically significant if the p-value was <0.05. Graphs were generated using Graphpad Prism version 9.4.

## Results

### *In vitro*, WC-rBS vaccine but not rCTB shows anti-inflammatory properties

As the immunomodulatory properties of CTB have been previously reported, we evaluated rCTB and the WC-rBS vaccine (based on rCTB and *Vibrio cholerae* strains) in an endotoxin tolerance experiment with the goal to get a first sense of their anti-inflammatory properties. This consisted of the treatment of immune cells (THP1 or PBMC) with compounds of interest before induction of an inflammatory environment with a high dose of *E. coli* LPS. Surprisingly, pretreatment with WC-rBS, but not with rCTB, significantly decreased secretion of the pro-inflammatory cytokine TNFα in comparison to the untreated group (Media). This was observed for both human THP-1 monocytes (Fig 1A) and human PBMCs (Fig 1B). In addition, WC-rBS vaccine exhibited a trend of increased secretion of the anti-inflammatory IL-10 cytokine by THP1 cells (Fig 1C). This trend was confirmed on PBMCs with significantly higher IL-10 secreted when cells are previously treated with WC-rBS (Fig 1D). In contrast, rCTB failed to promote IL-10 secretion from either PBMCs or THP1 cells. These results suggested that, unlike whole WC-rBS vaccine, highly purified rCTB alone is not sufficient to promote an anti-inflammatory response.

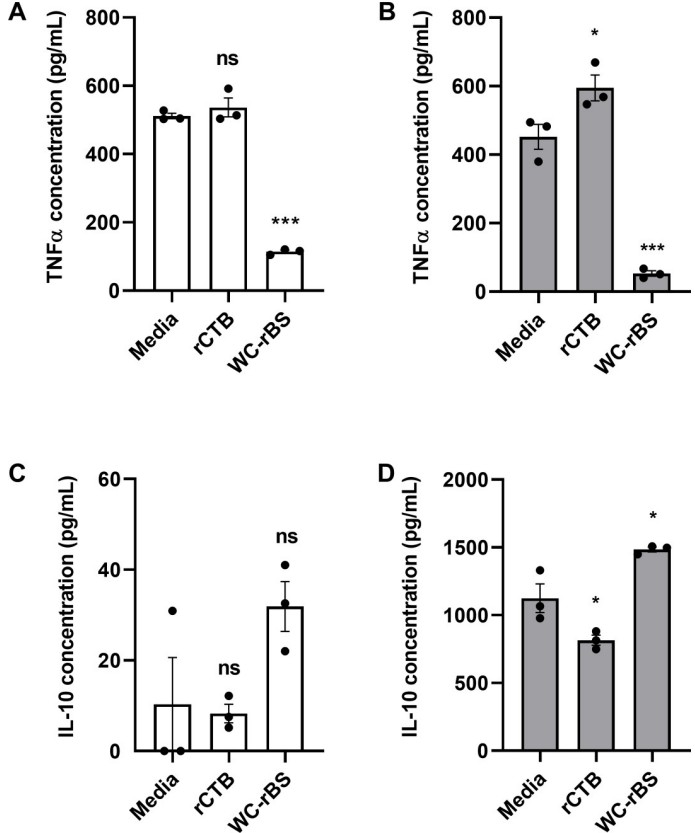

**Fig 1. Suppression of TNFα response and enhancement of IL-10 response to LPS in THP-1 monocytes or PBMCs pretreated with WC-rBS vaccine.** THP-1 monocytes (A, C) or PBMCs (B, D) were pretreated for 16 to 24 h with media, rCTB (10 μg/mL), or WC-rBS vaccine (accordingly diluted to 10μg/mL rCTB) and then challenged with LPS (1 μg/mL). Supernatants were collected 6h post challenge to assess TNFα secretion from PBMCs (B) or 24 h post-challenge to assess TNFα (A) secretion from THP-1 cells and IL-10 secretion from THP-1 cells (C) and PBMCs (D). Bars represent mean ±SEM. Statistics: One-way ANOVA with Dunnett's multiple comparison. ns: non-significant, * p-value < 0.05, *** p-value < 0.001.

## WC-rBS vaccine shows anti-inflammatory properties *in vivo*

Given the anti-inflammatory properties of WC-rBS *in vitro*, it was then tested in a DSS colitis mouse model. In the first experiment, two different doses of WC-rBS (corresponding to WC-rBS dilutions to achieve 25 or 5 μg rCTB per dose) were administrated five times every three days before, during and after 5 days of inflammatory 2,5% DSS treatment (Fig 2A) and compared to mesalamine drug given *ad libitum* for the duration of the experiment. Mesalamine, also known as mesalazine, is an approved 5-aminosalicylic acid medication used to treat IBD in humans. Mice were sacrificed on day 12, 7 days after the administration of DSS, which represented the phase of wound-healing / beginning of the chronic colitis phase.

The DSS + mesalamine group validated the model with a significantly lower body weight loss over time (Fig 2B), improved DAI at day 12 (Fig 2C), improved colon length at day 12 (Fig 2D) and improved global histological score (Fig 2E) on day 12 when compared to the untreated control group (DSS + PBS). WC-rBS 5 also exhibited anti-inflammatory effects, showing a decreased DAI score at 3.11 ± 0.26, close to significance (p-value = 0.054) in comparison to the DSS + PBS control group DAI at 4.00 ± 0.30 (Fig 2C). Although not significant, the body weight loss was also improved at the end of the experiment (Fig 2B). This correlates

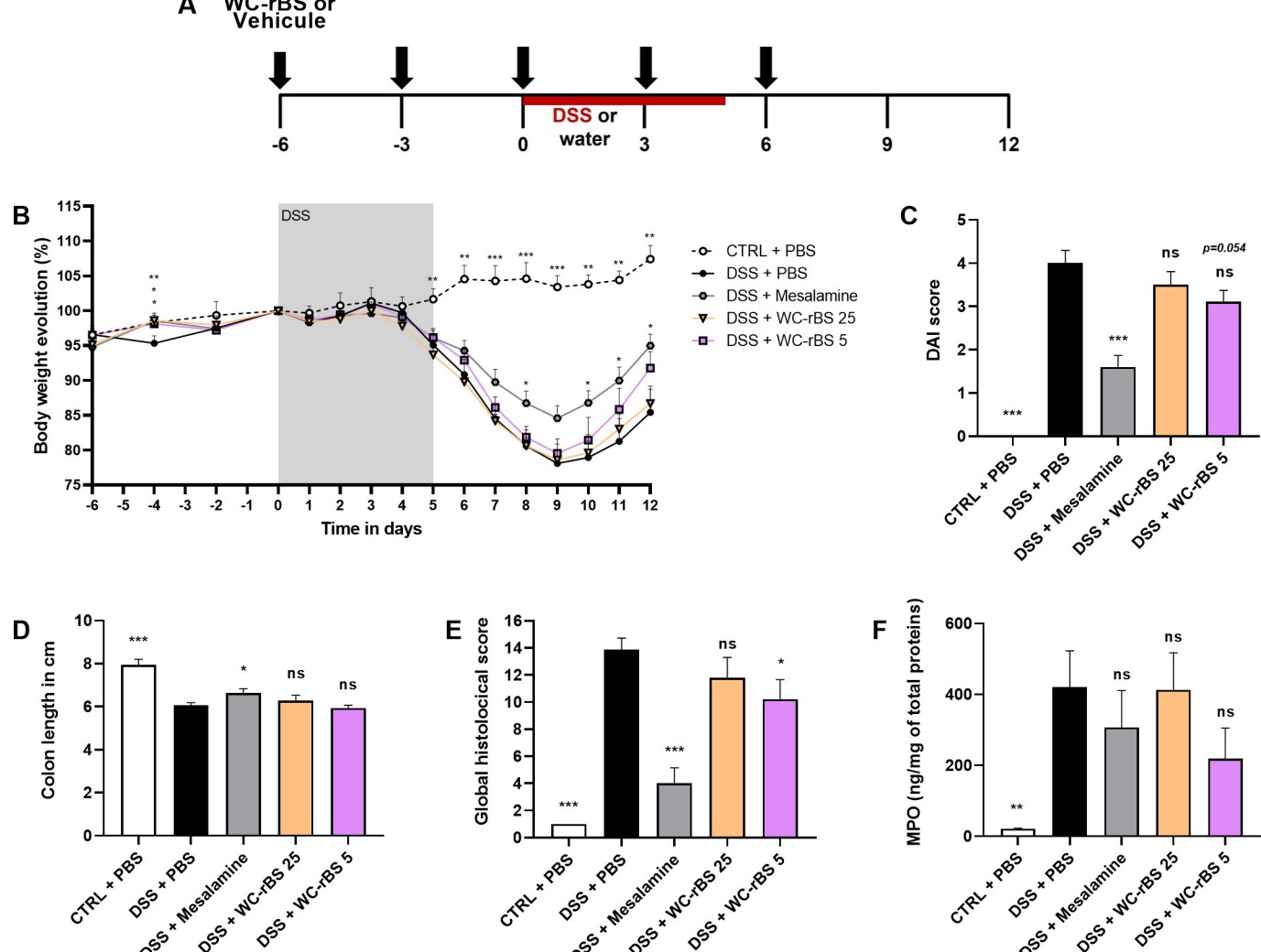

**Fig 2. Anti-inflammatory potential of WC-rBS vaccine in a colitis DSS model.** Mice were exposed to 2.5% DSS for 5 consecutive days (days 0 to 5) and orally administrated with WC-rBS 25, WC-rBS 5, or PBS on days -6, -3, 0, 3 and 6 (A). As a control, one group of mice received Mesalamine drug *ad libitum* over the whole experiment. One healthy control group (CTRL + PBS) was not exposed to DSS. Body weight was recorded individually throughout the entire experiment and percent change calculated (B). On day 12 mice were euthanized. DAI score(C), colon length (D) and global histological score (E) were determined and MPO quantified by ELISA on colonic sections (F). Bars represent mean ±SEM. Statistics: Permutation test. ns: non-significant, * p-value < 0.05, ** p-value < 0.01, *** p-value < 0.001.

to a significant decrease in inflammation at the histological level for WC-rBS 5 (10.22 ± 1.43) in comparison to the DSS + PBS control group (13.89 ± 0.84; Fig 2E). WC-rBS 5 also tended to decrease the level of inflammatory cell infiltrates in the colon measured by quantifying the protein level of MPO. On the other hand, WC-rBS 25 did not demonstrate efficacy in day 12 readouts.

## Decreasing WC-rBS dose improve its immunomodulatory properties

Given the observation that WC-rBS tended to reduce colitis disease activity in Fig 2, we hypothesized that lower doses of WC-rBS may be more effective in inducing a tolerogenic state, as was reported for CTB-KDEL [18, 22]. We performed a dose titration study to assess lower WC-rBS vaccine doses (corresponding to WC-rBS dilutions to achieve 5, 3, or 1 µg

rCTB per dose). Experimental design was identical to the first experiment (Fig 3A). As in Fig 2, the reduction in disease activity in the DSS + mesalamine group compared to untreated mice (DSS only) validated the model. Mice from this group had lower DAIs (Fig 3C) and lower histological scores (Fig 3D). However, it is important to note the global lower inflammatory level induced by DSS in this experiment. For all parameters evaluated, DSS + PBS group showed lower scores in comparison to the first *in vivo* experiment (Fig 2).

As predicted, disease outcomes were inversely WC-rBS dose dependent with WC-rBS 1 inducing the most disease improvement on day 12. Both WC-rBS 3 and WC-rBS 1 had significantly lower DAI, 1.30 ± 0.30 and 0.70 ± 0.26 respectively, in comparison to the control group, (2.60 ± 0.34, Fig 3C). This corresponded to a DAI improvement of 50 and 73% respectively. Although none of the treated groups showed reduced body weight loss in comparison to the DSS+PBS control group (Fig 3B), the inverse WC-rBS dose dependance was observed in body weight evolution data from day 7 to day 12 amongst treatment groups. WC-rBS 1 group weight loss was lower than for WC-rBS 3 and WC-rBS 5 groups. The individual scores that composed DAI are available in S1 Fig.

In keeping with the improved day 12 DAI, a significant improvement of colon length (Fig 3D) and a significant decrease in disease activity at the histological level (Fig 3E) was observed for all WC-rBS groups on day 12. The lowest dose presented the highest improvement with colon lengths at 7.09 cm ± 0.19 for WC-rBS 5, 7,32 cm ± 0.15 for WC-rBS 3 and 7,78 cm ± 0.14 for WC-rBS 1 versus 6,45 cm ± 0.13 for the DSS + PBS control group (Fig 3D). To note, the WC-rBS 1 group presented a colon length close to the normal (colon length in the control + PBS group at 8,00 cm ± 0.22). Similarly, WC-rBS 1 showed the highest improvement at histological level with global histological score at 2.90 ± 1.27 for WC-rBS 5, 1.90 ± 0.90 for WC-rBS 3 and 1.00 ± 0.00 for WC-rBS 1 versus 8.40 ± 1.44 for the DSS + PBS control group (Fig 3E). The inverse WC-rBS dose response was observed for 4 out of the 5 parameters scored to evaluate the global histological disease activity, including the inflammation severity, extent, regeneration and crypt damage scores (S1D–S1G Fig). All factors were significantly reduced for both WC-rBS 3 and WC-rBS 1 in comparison to the DSS + PBS group. Only inflammation severity and extent scores showed significant reduction for the WC-rBS 5 group (S1D and S1E Fig). To note, the protective effects exerted by WC-rBS were greater than those induced by the positive control group which received the anti-inflammatory drug mesalamine. Representative histological images shown for each group (Fig 4A and 4B) highlight the anti-inflammatory effect observed on day 12 in the WC-rBS treated groups among the different studied parameters (S1D–S1G Fig). No significant change was detected in MPO levels across all treatment groups (Fig 3F). Overall, these data confirm the safety and the potential of WC-rBS vaccine in a mouse model of colitis, on both the macroscopic and tissue levels.

## WC-rBS vaccine potential when administrated in a curative mode

In the same experiment, we also administered WC-rBS in a curative mode, only after the beginning of the DSS treatment with two doses of WC-rBS on days 3 and 6 (Fig 5A). Both WC-rBS 5 and WC-rBS 1 groups significantly improved by 50% the DAI score in comparison to the DSS + PBS control group with DAI score of 1.30 ± 0.33 for WC-rBS 5, 1.30 ± 0.30 for WC-rBS 1 and 2.60 ± 0.34 for the DSS control group (Fig 5C). At the end of the experiment, although body weight ratio was not improved for WC-rBS groups, the trends of improved stool consistency and decreased presence of occult blood explained the significantly reduced DAI score (S2A–S2C Fig). In addition, the colon length was significantly improved for WC-rBS 5 (7.74 cm ± 0,16) and WC-rBS 1 (7,85 cm ± 0.27) whereas the DSS + PBS group presented a score at 6,45 cm ± 0.13 (Fig 5D). In comparison to the non DSS treated group, WC-rBS

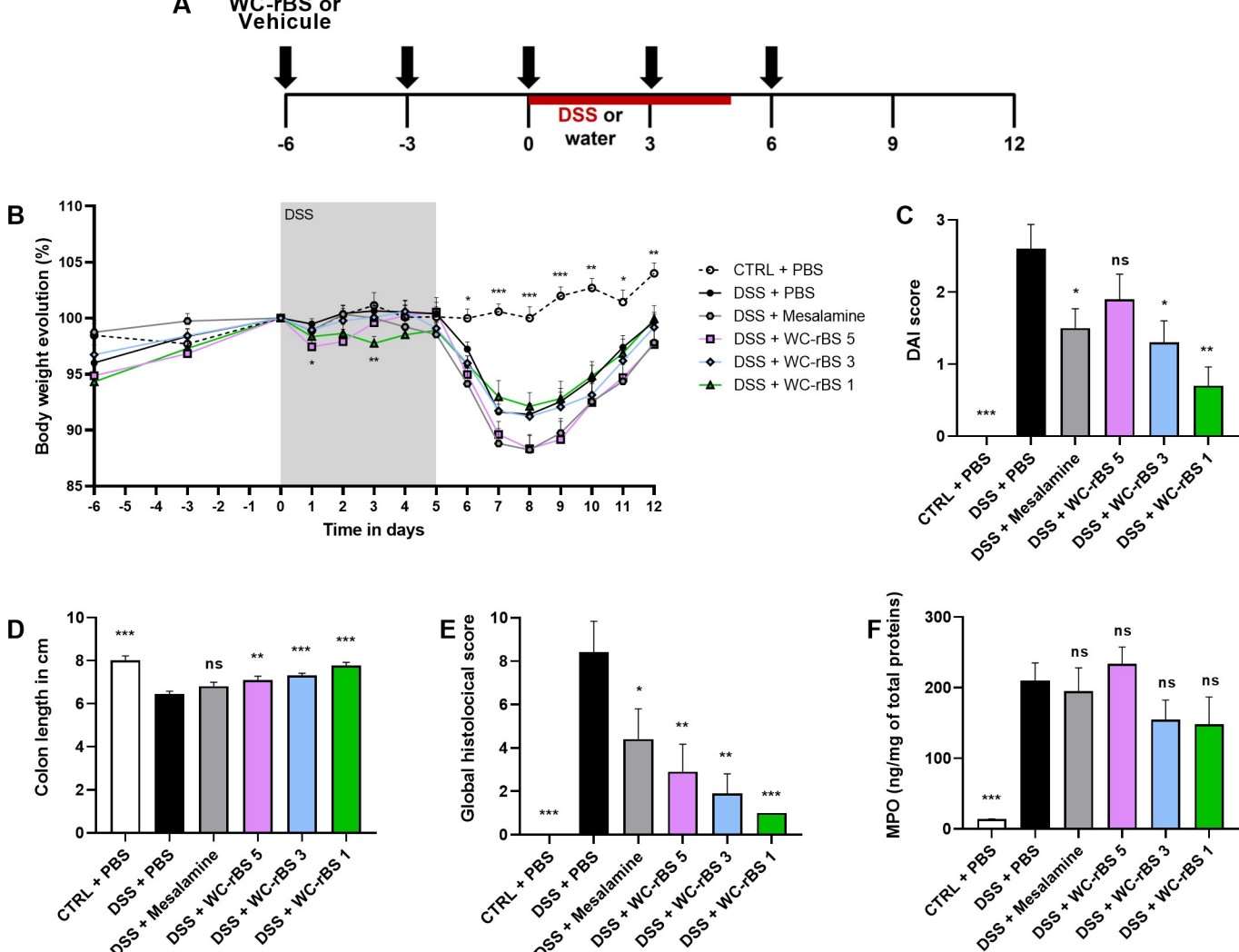

**Fig 3. Lower WC-rBS vaccine doses improved DSS colitis outcome.** Mice were exposed to 2.5% DSS for 5 consecutive days (days 0 to 5) and orally administrated with WC-rBS 5, WC-rBS 3, WC-rBS 1 or PBS on days -6, -3, 0, 3 and 6 (A). As a control, one group of mice received Mesalamine drug *ad libitum* over the whole experiment. One healthy control group (CTRL + PBS) was not exposed to DSS. Body weight was recorded individually throughout the entire experiment and percent change calculated (B). On day 12 mice were euthanized. DAI score (C), colon length (D) and global histological score (E) were determined and MPO quantified by ELISA on colonic sections (F). Bars represent mean ±SEM. Statistics: Permutation test. ns: non-significant, * p-value < 0.05, ** p-value < 0.01, *** p-value < 0.001.

groups presented a colon length close to the normal (colon length in the control + PBS group at 8,00 cm ± 0.22). The global histological score was also significantly decreased for WC-rBS 5 (2.90 ± 1.69) and WC-rBS 1 (1.60 ± 0.31) whereas the DSS + PBS group presented a score at 8.40 ± 1.44 (Fig 5E). As observed with the WC-rBS administration in a preventive mode, inflammation severity, extent, regeneration and crypt damages scores were all significantly reduced after oral administration of WC-rBS 1 in a curative mode; whereas only inflammation severity, extent score showed significant reduction for the WC-rBS 5 group (S2D and S2E Fig). Representative histological images shown for each group (Fig 4A and 4C) highlight the anti-inflammatory effect observed on day 12 in the WC-rBS treated groups among the different studied parameters (S2D–S2G Fig). In addition, WC-rBS 1 significantly decreased the MPO

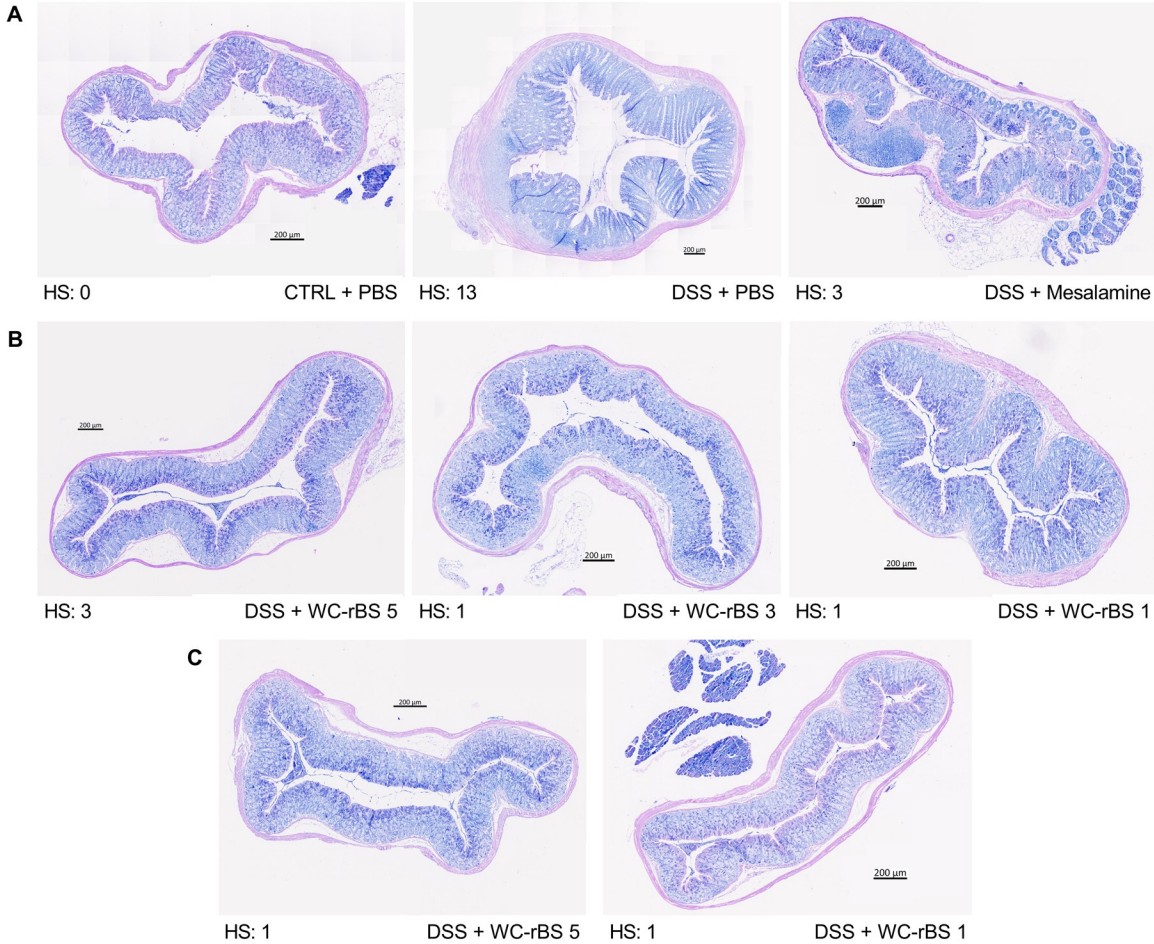

**Fig 4. WC-rBS vaccine doses improved histological damages.** Mice were exposed to 2.5% DSS for 5 consecutive days (0 to 5) and orally administered with WC-rBS 5, WC-rBS 3 or WC-rBS 1 as either preventive or curative treatments. Additional groups served as control groups, receiving no treatment nor DSS, or only DSS, or DSS plus ad libitum mesalamine drug. On day 12 mice were euthanized, sections of colonic tissues (4μm) were sampled and stained with May-Grunwald-Giemsa for histological evaluation of control groups (A), groups administered preventively with WC-rBS (B), and groups administered curatively with WC-rBS (C). Related histological scores are indicated below each picture.

level in colonic tissue reflecting a reduction in inflammatory neutrophil infiltration (Fig 5E). These data demonstrate the therapeutic benefit of WC-rBS in colitis.

## Discussion

WC-rBS vaccine is an approved vaccine administrated for the prevention of cholera gastroenteritis. Because of its unique composition of both rCTB and LPS from inactivated strains of *Vibrio cholerae*, we hypothesized WC-rBS may have a positive impact on inflammatory bowel disease. Initial *in vitro* endotoxin tolerance experiments showed the ability of WC-rBS vaccine to modulate cellular inflammation with decreased TNFα and increased IL-10 secretions by both human PBMCs and THP1 monocytes cells in an inflammatory environment after pre-exposure to the vaccine. On the other hand, rCTB alone did not present immunomodulatory properties in either the PBMC or the THP-1 monocyte endotoxin models. Although CTB/rCTB has previously been reported to be anti-inflammatory in such endotoxin tolerance models [6, 8], some discrepancies in the field suggest that the source of CTB may play a critical

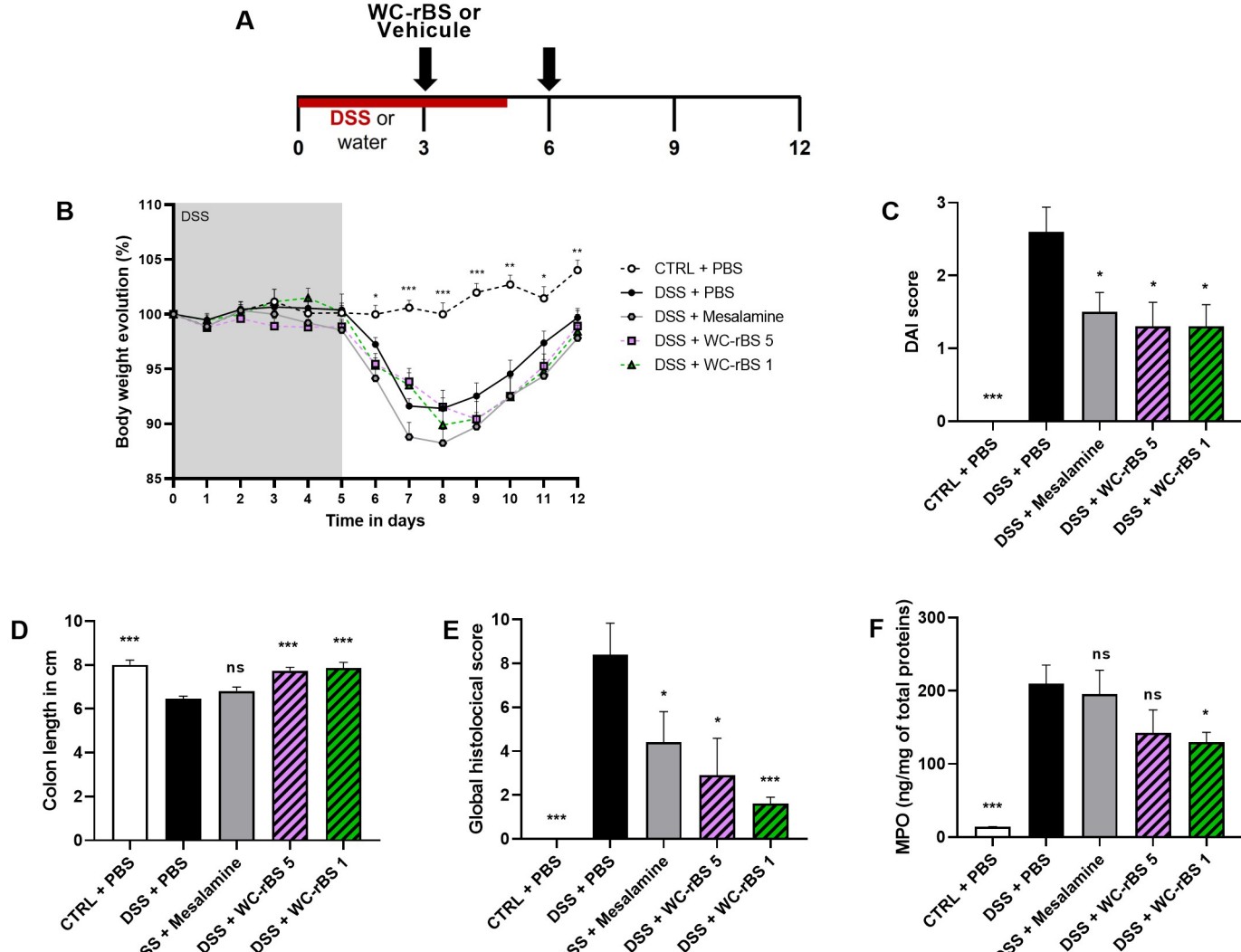

**Fig 5. WC-rBS vaccine also improved DSS colitis outcome when administered curatively.** Mice were exposed to 2.5% DSS for 5 consecutive days (0 to 5) and orally administrated with WC-rBS 5, WC-rBS 1, or PBS on days 3 and 6 (A). As controls, one group of mice received Mesalamine drug *ad libitum* over the whole experiment and one healthy control group was not exposed to DSS. Body weight was recorded individually over the entire experiment and percent change calculated (B). On day 12 mice were euthanized. DAI score (C), colon length (D) and global histological score (E) were determined and MPO quantified by ELISA on colonic sections (F). Bars represent mean ±SEM. Statistics: Permutation test. ns: non-significant, * p-value < 0.05, ** p-value < 0.01, *** p-value < 0.001.

role. Indeed, CTB can be produced recombinantly in many different production systems [6] or purified from whole cholera toxin and both products do not exert a similar effect. When purified from whole CT, residual CT subunit A was recently reported by Baldauf *et al.* to be anti-inflammatory in a endotoxin tolerance experiment on RAW264.7 murine macrophage cells, whereas endotoxin and CTA free rCTB was not [6]. To clarify CTB's effect in the experimental conditions used in Fig 1, purified CTB and rCTB were tested in the *in vitro* endotoxin tolerance model utilizing PBMCs. Interestingly, purified CTB but not rCTB was able to decrease TNFα (S3 Fig); in agreement with Baldauf et al [6].

Given the anti-inflammatory effects of WC-rBS in endotoxin tolerance models, the vaccine was then assessed *in vivo* in the DSS colitis mouse model. In two independent experiments, we demonstrated the safety of WC-rBS vaccine. Although one mouse died in the first *in vivo*

experiment due to severe colitis associated with high body weight loss overall there was no significant difference in mortality rate amongst the groups. This supports the clinical data reported by Dembiński *et al* showing that, while less immunogenic than in healthy children, WC-rBS vaccine is safe and well tolerated in IBD patients [23]. It should be noted that there was a difference in global inflammation level among the animal studies performed, despite employing the same experimental protocol. Indeed, comparison of the DSS + PBS control groups revealed high differences in terms of DAI and Global histological scores. Also, the maximum body weight loss observed varied from 10 to 20%. Despite being widely used and well described, the DSS colitis model is highly sensitive to environmental factors such as animals' microbiota, DSS source and dosing, etc [24]. In the *in vivo* studies presented here, the animals are the only source of variability between both experiments, suggesting a role in discrepancies observed. Similarly, other teams in the field using the same supplier facility recently reported similar issues at the same time with their respective DSS models (personal discussions).

Overall, in this study the immunomodulatory properties of WC-rBS vaccine reduced DSS colitis severity and demonstrated significant reduction of tissues damages. In a curative mode, administration of only two low doses of WC-rBS vaccine was as effective as the *ad libitum* administration of mesalamine benchmark drug over the entire experimental study in term of DAI. Taking into account the body weight loss observed, the beneficial effect of WC-rBS in the DSS model appears to be localized at the tissue level, possibly via a wound healing effect. Notably, at the histological level on day 12, WC-rBS vaccine was more effective than mesalamine. In addition, when administered preventively (5 doses), WC-rBS vaccine was superior to the mesalamine at reducing colitis with respect to both disease index and histological levels. The MPO level, quantified on day 12 does not always correlate with histological scores. However, as an indirect marker of inflammation, MPO levels are expected to be higher at the peak of inflammation on day 8 in the DSS model used. In addition, sampling several parts of the intestine for MPO quantification could give us a better overview. Finally, establishing WC-rBS mechanism of action would help to interpret those levels.

For a better understanding of WC-rBS vaccine mechanism of action, each component (rCTB and *V. cholerae* strains) should be individually evaluated. Additional studies to discern WC-rBS impact throughout the inflammatory period of this acute DSS colitis model are also required. Along with acute DSS, the chronic DSS mouse model and other animal models of IBD are available [25, 26] for deeper characterization of WC-rBS. Chemically induced, based on innate or adaptative immunity alteration or with intestinal epithelial defects, all models have specificities that can contribute to highlight WC-rBS vaccine potential in IBD.

Repurposing of drugs and vaccines for novel indications has emerged as a promising strategy to decrease risk and cost of drug development [27]. In the vaccine field, intravesical injections of Bacillus Calmette-Guérin (BCG) vaccine demonstrated lower recurrence and progression rates of bladder cancer and is currently the immunotherapy of choice [28]. Furthermore, in a 2017 study the meningococcal B vaccine demonstrated promising cross-protection against *Neisseria gonorrhoeae* with vaccinated people having significantly lower rates of gonorrheal disease than unvaccinated ones [29]. Repurposing the WC-rBS vaccine in IBD would have several advantages. WC-rBS safety has been demonstrated in IBD patients [23] as well as healthy subjects for more than 20 years. Further clinical development would require likely refinement including multiple administrations of WC-rBS vaccine, but time and cost could be saved by bypassing Phase 1 safety studies, especially if low WC-rBS vaccine doses are confirmed to be the most promising in the prevention and/or treatment of IBD. WC-rBS may be a unique opportunity to provide a safe, well-tolerated, oral therapy to complement the current standard-of-care and potentially improve the outcomes for IBD patients.

In summary, WC-rBS vaccine demonstrated a remarkable anti-inflammatory effect which could have a role in the prevention and/or the treatment of IBD in humans. Both preventive and curative administration of WC-rBS induced highly significant improvement in disease outcome and reduced colonic epithelial damage. This is, to our knowledge, the first report of a licensed prophylactic infectious disease vaccine exhibiting a therapeutic effect in an inflammatory autoimmune disease such as IBD. Because no single animal or experimental model completely recapitulates the clinical and histopathological characteristics of human IBD, the WC-rBS vaccine as well as its individual components, require further investigation to consolidate evidence reported thus far.

## Supporting information

**S1 Fig. Individual DAI and histological parameters evaluated after WC-rBS 5, WC-rBS 3, and WC-rBS 1 preventive administration in a DSS model.** Day 12 DAI and histological scores reported in Fig 3 are calculated as the sum of different parameters. DAI is the sum of the following parameters: the body weight ratio between day 12 and day 0: the first day of DSS administration (A), the stool consistency (B) and the presence of occult blood (C). The global histological score refers to the sum of the severity (D) and extent (E) of inflammation, the epithelium regeneration (F), the crypt damages (G) and the percentage of involvement (H). Bars represent mean ±SEM. Statistics: Permutation test. ns: non-significant, * p-value < 0.05, ** p-value < 0.01, *** p-value < 0.001.
(TIFF)

**S2 Fig. Individual DAI and histological parameters evaluated after WC-rBS 5 and WC-rBS 1 curative administration in a DSS model.** Day 12 DAI and histological scores reported in Fig 4 are calculated as the sum of different parameters. DAI is the sum of the following parameters: the body weight ratio between day 12 and day 0: the first day of DSS administration (A), the stool consistency (B) and the presence of occult blood (C). The global histological score refers to the sum of the severity (D) and extent (E) of inflammation, the epithelium regeneration (F), the crypt damages (G) and the percentage of involvement (H). Bars represent mean ±SEM. Statistics: Permutation test. ns: non-significant, * p-value < 0.05, ** p-value < 0.01, *** p-value < 0.001.
(TIFF)

**S3 Fig. Suppression of TNFα response to LPS in PBMCs pretreated with purified CTB but not rCTB.** PBMCs from 3 different healthy donors were pretreated for 16h with media, rCTB (10µg/mL), or purified CTB from CT (Sigma C9903, 10µg/mL) and then challenged with LPS (1µg/mL). Supernatants were collected 6h post challenge to assess TNFα concentration. Bars represent mean ±SEM. Statistics: One-way ANOVA with Dunnett's multiple comparison. ns: non-significant, *** p-value < 0.001.
(TIFF)

**S1 Data.**
(XLSX)

## Author Contributions

**Conceptualization:** Klaus Schwamborn, Melissa Hanson.

**Formal analysis:** Marine Meunier, Adrian Spillmann, Christel Rousseaux, Klaus Schwamborn, Melissa Hanson.

**Investigation:** Marine Meunier, Adrian Spillmann, Christel Rousseaux, Klaus Schwamborn, Melissa Hanson.

**Methodology:** Christel Rousseaux, Klaus Schwamborn, Melissa Hanson.

**Project administration:** Christel Rousseaux, Klaus Schwamborn, Melissa Hanson.

**Resources:** Christel Rousseaux.

**Supervision:** Klaus Schwamborn, Melissa Hanson.

**Validation:** Marine Meunier, Adrian Spillmann, Christel Rousseaux, Klaus Schwamborn, Melissa Hanson.

**Visualization:** Marine Meunier, Melissa Hanson.

**Writing – original draft:** Marine Meunier.

**Writing – review & editing:** Marine Meunier, Adrian Spillmann, Christel Rousseaux, Klaus Schwamborn, Melissa Hanson.

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
