## [Decision Letter · Decision Letter 0]

26 Jun 2023

PONE-D-23-07112An oral cholera vaccine in the prevention and/or treatment of inflammatory bowel diseasePLOS ONE

Dear Dr. Meunier,

Thank you for submitting your manuscript to PLOS ONE. After careful consideration, we feel that it has merit but does not fully meet PLOS ONE’s publication criteria as it currently stands. Therefore, we invite you to submit a revised version of the manuscript that addresses the points raised during the review process.

**Kindly find the comments of reviewer 1, on the bottom of this message, to be considered in your revised manuscript, **

We look forward to receiving your revised manuscript.

Kind regards,

Imad Al Kassaa

Academic Editor

PLOS ONE

Journal Requirements:

I have read the journal's policy and the authors of this manuscript have the following competing interests: Marine Meunier, Adrian Spillmann, Klaus Schwamborn and Melissa Hanson were all employees of the Valneva group when the work was done. The Swedish subsidiary of the Valneva group owns Dukoral®, i.e. the WC-rBS vaccine.

Reviewers' comments:

Reviewer's Responses to Questions

**Comments to the Author**

1. Is the manuscript technically sound, and do the data support the conclusions?

Reviewer #1: Yes

2. Has the statistical analysis been performed appropriately and rigorously? 

Reviewer #1: Yes

3. Have the authors made all data underlying the findings in their manuscript fully available?

Reviewer #1: Yes

4. Is the manuscript presented in an intelligible fashion and written in standard English?

Reviewer #1: Yes

5. Review Comments to the Author

Reviewer #1: The manuscript entitled " An oral cholera vaccine in the prevention and/or treatment of inflammatory bowel disease" by Meunier et al. describes studies aimed to investigate the potential beneficial 101 properties of WC-rBS vaccine in autoimmune disease. In general, the subject is interesting and it is the first report of the therapeutic effect of WC-rBS – a prophylactic oral cholera vaccine – for the treatment of intestinal colitis. This paper is clear and straightforward, the parameters asserted are clear and the data well represented. However, there are some problems in this study should be considered before this manuscript being accepted.

Comments:

Page 9 L49: E. coli should be in italic … You should review all bacterial names.

Page 10 L79: Before start use the abbreviation of V. cholerae, you should mention before the use of this abbreviation “Vibrio cholerae (V. cholerae)”.

Page 11 L91: The author should correct: TGF-β.

Page 11 L108: “To our knowledge, this is the first report of the therapeutic effect of WC-rBS – a prophylactic oral cholera vaccine – for the treatment of intestinal colitis” … Better to say for the treatment and prevention of intestinal colitis.

Page 11: Add a reference for the protocol of isolation and use of PBMC

Figure 3: Please add the protocol like Figure 2 and 3

In the PBMC and TPH-1 in vitro experiment, I recommend to test another cytokine mainly IL-12, INF-γ and calculate the IL-10/IL-12 ratio.

In the DSS in vivo experiment, auteur should add a representative histological section for each group and it is very important to add the Shortening of colon length if available and test different genes expression like CXCL2, IL-6 for inflammation and ZO-1, occludine from colonic samples.

In the DSS in vivo experiment, why you orally administrate WC-rBS on days -6, -3, 0, 3 and 6 not every day? And why the mice were exposed to 2.5% DSS not for example 3.5% or 5%?

Why the auteur uses DSS colitis mouse model not TNBS colitis mouse model?

6. PLOS authors have the option to publish the peer review history of their article (what does this mean?). If published, this will include your full peer review and any attached files.

Reviewer #1: **Yes: **Mazen zaylaa

---

## [Author Response · Author response to Decision Letter 0]

2 Aug 2023

Reviewers' Comments to the Authors:

The manuscript entitled " An oral cholera vaccine in the prevention and/or treatment of inflammatory bowel disease" by Meunier et al. describes studies aimed to investigate the potential beneficial properties of WC-rBS vaccine in autoimmune disease. In general, the subject is interesting and it is the first report of the therapeutic effect of WC-rBS – a prophylactic oral cholera vaccine – for the treatment of intestinal colitis. This paper is clear and straightforward, the parameters asserted are clear and the data well represented. However, there are some problems in this study should be considered before this manuscript being accepted. 

Thank you. 

Comments: 

Page 9 L49: E. coli should be in italic … You should review all bacterial names. � This has been checked and modified (Page 3 L49).

Page 10 L79: Before start use the abbreviation of V. cholerae, you should mention before the use of this abbreviation “Vibrio cholerae (V. cholerae)”. � “(V. cholerae)” abbreviation has been added L50 (Page 3 L50).

Page 11 L91: The author should correct: TGF-β. � This has been corrected (Page 5 L91).

Page 11 L108: “To our knowledge, this is the first report of the therapeutic effect of WC-rBS – a prophylactic oral cholera vaccine – for the treatment of intestinal colitis” … Better to say for the treatment and prevention of intestinal colitis. � This has been corrected (Page 5 L108). 

Page 11: Add a reference for the protocol of isolation and use of PBMC. � Burkart et al reference has been added L120: “(protocol adapted from Burkart et al (8)” (Page 6 L120). 

Figure 3: Please add the protocol like Figure 2 and 3 To be finalized, change fig ref in the text, check format � Fig 3 has been modified to include the protocol. The text has been checked for figure references, as well as the figure caption and format. 

In the PBMC and TPH-1 in vitro experiment, I recommend to test another cytokine mainly IL-12, INF-γ and calculate the IL-10/IL-12 ratio. � Indeed, studying multiple cytokine expressions would be interesting to understand better the pro/anti-inflammatory profile of the cells. During our assay development on THP-1 and PBMC, both IFN-y and IL-12 have been assessed. However, under the tested conditions, we were not able to detect those two cytokines. For this reason, we decided to focus in vitro on TNF-α and IL-10 to receive first insights of the anti-inflammatory properties of WC-rBS.

In the DSS in vivo experiment, auteur should add a representative histological section for each group and it is very important to add the Shortening of colon length if available and test different genes expression like CXCL2, IL-6 for inflammation and ZO-1, occludine from colonic samples. � An additional Figure has been created (Fig. 4): It shows representative histological section for each experimental group from the in vivo study in which mice were exposed to 2.5% DSS for 5 consecutive days (0 to 5) and orally administered with WC-rBS 5, WC-rBS 3 or WC-rBS 1 as either preventive or curative treatments. Additional groups served as control groups, receiving no treatment nor DSS, or only DSS, or DSS plus ad libitum mesalamine drug. Histological score are also mentioned on Fig. 4. Figure caption has been added, and comments on this figure added to the manuscript text.

For each in vivo experiment, length of colon data has been added to respective Figure (Fig. 2, Fig. 3 and Fig. 5), panel C. Figures caption and numbers were updated, and comments on this parameter added to the manuscript text. 

Concerning the measurement of additional gene expression, we evaluated IL-6 on day 12 (7 days after the last DSS administration) for all experiments. Unfortunately, no significant differences were observed between groups. This could be explained by the fact that in the DSS model, chronic‐like inflammation occurs on Day 12 with low levels of inflammatory cytokines (IL-6, CXCL2, …) expressed (in comparison to the acute phase occurring at Day 8). Evaluate those genes expression earlier, during the acute phase of inflammation could be interesting for a better understanding of the WC-rBS anti-inflammatory mechanism of action, especially on day 8 at the peak of inflammation. In addition, ZO-1 and occludine are both tight junction proteins studied for the evaluation of intestinal permeability. At this stage of the WC-rBS evaluation, we only focused on its anti-inflammatory potential. Indeed, those two proteins gene expression could be considered for further investigations of WC-rBS potential. 

In the DSS in vivo experiment, why you orally administrate WC-rBS on days -6, -3, 0, 3 and 6 not every day? And why the mice were exposed to 2.5% DSS not for example 3.5% or 5%? 

Different timings were considered for the oral administration of WC-rBS. Based on bibliography, we decided to follow a 3 days interval timing as described by Baldauf et al, 2017 and Royal et al, 2019. In our case, since WC-rBS is a prophylactic vaccine, we decided to start administration before the DSS treatment as a first step of our investigations. Then, we also evaluate its administration in a curative mode. In addition, to avoid animals stress due to handling (WC-rBS being administrated by oral gavage), we considered the daily administration not appropriate for our studies. 

At Intestinal Biotech Development, the DSS colitis mouse model uses a dosage at 2,5% as described almost 20 years ago by Melgar et al, 2005. This model is described as a reproducible model of acute colitis that induce no or low mortality. In addition, Intestinal Biotech Development is a CRO having a very strong track record in selecting, performing in colitis mouse models. This is why it has been decided to follow the established model. 

References: 

• Baldauf KJ, Royal JM, Kouokam JC, Haribabu B, Jala VR, Yaddanapudi K et al. Oral administration of a recombinant cholera toxin B subunit promotes mucosal healing in the colon. Mucosal Immunol 2017; 10(4):887–900.

• Royal JM, Reeves MA, Matoba N. Repeated Oral Administration of a KDEL-tagged Recombinant Cholera Toxin B Subunit Effectively Mitigates DSS Colitis Despite a Robust Immunogenic Response. Toxins (Basel) 2019; 11(12).

• Melgar S, Karlsson A, Michaëlsson E. Acute colitis induced by dextran sulfate sodium progresses to chronicity in C57BL/6 but not in BALB/c mice: correlation between symptoms and inflammation. Am J Physiol Gastrointest Liver Physiol 2005; 288(6):G1328-38.

Why the auteur uses DSS colitis mouse model not TNBS colitis mouse model? � Both DSS and TNBS models were considered for the in vivo evaluation of WC-rBS and its anti-inflammatory potential. DSS model of colitis is a validated animal model described to evaluate and confirm the anti‐inflammatory properties of drugs in IBD. It is also well suitable for the evaluation of the innate immune response and for the evaluation of factors maintaining or reestablishing epithelium integrity. Based on its composition, it was hypothesized WC-rBS could have a role on the improvement of disease outcome on both the macro- and tissue-level scales. DSS induces prominent diarrhea followed by a more superficial inflammation when compared to TNBS colitis in the colon. Indeed, TNBS model is described to be more invasive resulting in spreader inflammation and transmural colitis and more adapted for the evaluation of Crohn’s disease immunologic aspects. For these reasons, we determined the DSS colitis mouse model to be the most appropriate model to explore in the first place in our experiments.

Journal requirement to the Authors:

1. Please ensure that your manuscript meets PLOS ONE's style requirements, including those for file naming � This has been checked

2. We suggest you thoroughly copyedit your manuscript for language usage, spelling, and grammar. Upon resubmission, please provide the following: The name of the colleague or the details of the professional service that edited your manuscript � All authors reviewed the manuscript. In particular, Melissa Hanson, USA native speaker revised the manuscript for language usage, spelling and grammar.

Please confirm that this does not alter your adherence to all PLOS ONE policies on sharing data and materials, by including the following statement: "This does not alter our adherence to PLOS ONE policies on sharing data and materials.” � Authors confirm competing interest does not alter adherence to PlosOne policies. The new competing interest statement is: “Marine Meunier, Adrian Spillmann, Klaus Schwamborn and Melissa Hanson were all employees of the Valneva group when the work was done. The Swedish subsidiary of the Valneva group owns Dukoral®, i.e. the WC-rBS vaccine. This does not alter our adherence to PLOS ONE policies on sharing data and materials.”

Please include your updated Competing Interests statement in your cover letter; we will change the online submission form on your behalf. � Please see above

4. We note that you have stated that you will provide repository information for your data at acceptance. Should your manuscript be accepted for publication, we will hold it until you provide the relevant accession numbers or DOIs necessary to access your data. If you wish to make changes to your Data Availability statement, please describe these changes in your cover letter and we will update your Data Availability statement to reflect the information you provide. � We confirm that all relevant data are within the manuscript and its Supporting Information files. 

5. We note that you have included the phrase “data not shown” in your manuscript. Unfortunately, this does not meet our data sharing requirements. � The sentence referring to the “data not shown” has been removed for clarification of the text (Page 11, L236-237). 

6. Please review your reference list to ensure that it is complete and correct. � This has been checked and we didn’t make any edit of the reference list.

---

## [Editor Report · Decision Letter 1]

14 Aug 2023

An oral cholera vaccine in the prevention and/or treatment of inflammatory bowel disease

PONE-D-23-07112R1

Dear Dr. Meunier,

We’re pleased to inform you that your manuscript has been judged scientifically suitable for publication and will be formally accepted for publication once it meets all outstanding technical requirements.

Kind regards,

Imad Al Kassaa

Academic Editor

PLOS ONE
---

## [Editor Report · Acceptance letter]

18 Aug 2023

PONE-D-23-07112R1 

An oral cholera vaccine in the prevention and/or treatment of inflammatory bowel disease 

Dear Dr. Meunier:

I'm pleased to inform you that your manuscript has been deemed suitable for publication in PLOS ONE. Congratulations! Your manuscript is now with our production department. 

Kind regards, 

on behalf of

Professor Imad Al Kassaa 

Academic Editor

PLOS ONE